# Atomically engineering activation sites onto metallic 1T-MoS$_2$ catalysts for enhanced electrochemical hydrogen evolution

Yichao Huang[1,2], Yuanhui Sun[3], Xueli Zheng [4], Toshihiro Aoki [5], Brian Pattengale[6], Jier Huang[6], Xin He[3], Wei Bian[1], Sabrina Younan[2], Nicholas Williams[2], Jun Hu[1], Jingxuan Ge[1], Ning Pu[7], Xingxu Yan[8], Xiaoqing Pan[5,8,9], Lijun Zhang [3], Yongge Wei[1] & Jing Gu[2]

Engineering catalytic sites at the atomic level provides an opportunity to understand the catalyst's active sites, which is vital to the development of improved catalysts. Here we show a reliable and tunable polyoxometalate template-based synthetic strategy to atomically engineer metal doping sites onto metallic 1T-MoS$_2$, using Anderson-type polyoxometalates as precursors. Benefiting from engineering nickel and oxygen atoms, the optimized electro-catalyst shows great enhancement in the hydrogen evolution reaction with a positive onset potential of ~ 0 V and a low overpotential of −46 mV in alkaline electrolyte, comparable to platinum-based catalysts. First-principles calculations reveal co-doping nickel and oxygen into 1T-MoS$_2$ assists the process of water dissociation and hydrogen generation from their intermediate states. This research will expand on the ability to improve the activities of various catalysts by precisely engineering atomic activation sites to achieve significant electronic modulations and improve atomic utilization efficiencies.

[1] Key Lab of Organic Optoelectronics & Molecular Engineering of Ministry of Education, Department of Chemistry, Tsinghua University, Beijing 100084, P. R. China. [2] Department of Chemistry and Biochemistry, San Diego State University, 5500 Campanile Drive, San Diego, CA 92182-1030, USA. [3] State Key Laboratory of Superhard Materials, Key Laboratory of Automobile Materials of MOE, and School of Materials Science and Engineering, Jilin University, Changchun 130012, P. R. China. [4] Department of Materials Science and Engineering, Stanford University, Stanford, CA 94305, USA. [5] UC Irvine Materials Research Institute (IMRI), University of California - Irvine, Irvine, CA 92697, USA. [6] Department of Chemistry, Marquette University, Milwaukee, WI 53201-1881, USA. [7] Collaborative Innovation Center of Advanced Nuclear Energy Technology, Institute of Nuclear and New Energy Technology, Tsinghua University, Beijing 100084, P.R. China. [8] Department of Materials Science and Engineering, University of California - Irvine, Irvine, CA 92697, USA. [9] Department of Physics and Astronomy, University of California - Irvine, Irvine, CA 92697, USA. Correspondence and requests for materials should be addressed to L.Z. (email: lijun_zhang@jlu.edu.cn) or to Y.W. (email: yonggewei@mail.tsinghua.edu.cn) or to J.G. (email: jgu@sdsu.edu)

Electrocatalysis plays a crucial role in clean energy conversion, enabling the realization of sustainable paths for various commercial processes, such as hydrogen evolution[1,2]. Hydrogen ($H_2$) is an attractive energy carrier that may be utilized to harness green energy through combustion or electricity generation[3,4]. Electrochemically producing $H_2$ from renewable energy sources allows it to be generated in a more sustainable and locally distributed manner, eliminating transportation costs accrued from large-scale, centralized steam reforming[5,6]. However, there are obstacles that need to be overcome, such as developing electrocatalysts that are low cost, highly efficient, and maintain their stability over time[7–9]. Currently, efforts have been devoted to developing non-noble electrocatalysts to reduce cost issues, including carbon materials[9–11], transition metal carbides and nitrides[12,13], oxides[14,15], phosphides[16–19], and sulfides[20–22]. Many common heterogeneous catalytic synthetic strategies, like electrochemical deposition[23], chemical vapor deposition[24], or previous hydrothermal conversion methods[25–27], cannot effectively control catalyst activation sites. In these techniques, heteroatoms tend to form separate phases with different chemical components, rather than exhibiting uniform chemical doping. This causes the active sites to be unclear and limited to the interfaces[7,23,28] or edges[24,29]. Recently, more strategies have been developed to effectively improve active sites of 2D catalysts for the hydrogen evolution reaction (HER)[30], but it is still challenging to identify a means to disperse each catalyst's activation site uniformly and enable confident control of the metal activation sites at the atomic level. Here we demonstrate a reliable and tunable synthetic strategy derived from converting polyoxometalates (POMs), a special class of metal oxide anion nanoclusters with diverse chemical properties, physical properties, and well-defined structures[31–33]. This method accomplishes engineering highly conductive 1T-$MoS_2$ nanosheets with control of chemical doping at the atomic level, simultaneously.

In the continuous search for earth-abundant catalysts, $MoS_2$ serves as a role-model in research regarding catalysis for hydrogen evolution. For decades, its activity has been considered limited, due to the extremely high hydrogen adsorption free energy on $MoS_2$'s basal plane ($\Delta G_H = 1.92$ eV). This changed when theoretical calculations revealed the extremely thermo-neutral $\Delta G_H$ (0.08 eV) existing on $MoS_2$'s edge sites[34–37]. Since then, various strategies to engineer nanostructures of 2H-phase $MoS_2$-based materials[38,39] and increase the number of exposed edge sites have developed, such as fine-tuning the sulfur vacancies[20,40,41], changing conductive supports[38,42–44], and incorporating transition metal heteroatoms[25,45–48]. Unfortunately, 2H-$MoS_2$ applications in catalytic HER are limited by issues with density, active site reactivity, poor charge transport properties between layers of 2H-$MoS_2$, and inefficient electrical contact with its conductive support[43,44]. Alternatively, metallic 1T-phase $MoS_2$ (1T-$MoS_2$) exhibits more facile charge transport properties, thus enabling exposure of a larger quantity of active sites, which in turn produces superior HER performance[44,49,50]. Although these properties are attractive, they are impeded by the metastable tendencies of 1T-$MoS_2$ which allow the structure to easily revert back to 2H-$MoS_2$ through the intra-layer atomic plane glide[51,52]. Despite the capability of lithium-ion intercalations with lithium foil, organo-lithium, or lithium borohydride[50,53] to drive $MoS_2$ structure transformations from the hexagonal 2H-phase to the octahedral 1T-phase, large-scale applications are constrained due to consistent low yields reported by most synthetic strategies for metallic 1T-$MoS_2$ synthesis[44,49–53].

Recently, a hydrothermal process incorporating organic sulfur sources into $(NH_4)_6Mo_7O_{24}\cdot 4H_2O$ (denoted as $Mo_7$) has proven to be an efficient method to obtain highly purified and stable metallic 1T-$MoS_2$[49]. $Mo_7$ is a precursor belonging to the $\beta$-isomer

of Anderson-type POMs, a butterfly-shaped metal oxide cluster[54,55]. They are well-defined, nanostructured, 1:6 heteropolyanion clusters, composed of a single metal heteroatom $XO_6$ octahedron (X = Fe, Co, Ni, et al.) with six edge-sharing $MO_6$ (M = Mo, W) octahedrons[55]. This unique structure provides an opportunity to fine-tune the chemical environment of 1T-$MoS_2$ with various heteroatoms. With the hypothesis that Anderson-type POMs have the potential to be ideal precursors for 1T-$MoS_2$ production, a series of Anderson-type POM nanoclusters, $[XH_6Mo_6O_{24}]^{n-}$ (denoted as $XMo_6$; X = Fe$^{III}$, Co$^{III}$, n = 3; X = Ni$^{II}$, n = 4), are prepared as precursors for metallic 1T-$MoS_2$ nanosheets co-doped with a sequence of first row transition metal and oxygen atoms, with results indicating ultrathin metallic 1T-$MoS_2$ nanosheets may be achieved by this exclusive conversion method.

Though 1T-$MoS_2$ is a promising candidate for HER under acidic conditions, it suffers from a high overpotential under alkaline conditions, due to the slow water dissociation process[48]. Alkaline solution conditions are optimal because they allow both water oxidation and reduction to occur under the same conditions, which eliminates pH-induced energy and ion exchange consumption. The optimized Ni and O co-doped metallic 1T-$MoS_2$ (NiO@1T-$MoS_2$) nanosheet catalyst produced an extremely low overpotential of −46 mV with 10 mA cm$^{-2}$ current density for HER in a strong basic electrolyte (1.0 M KOH), comparable to the 20% Pt-based catalyst. Moreover, first-principles calculations suggest the kinetics of initial water dissociation and final hydrogen generation may effectively be promoted by doping both Ni and O atoms onto the basal planes of 1T-$MoS_2$. When designing catalysts, two major strategies exist that improve their activity: increasing the number of active sites and increasing the intrinsic activity of each existing active site. The research discussed herein employs one simple design principle to accomplish both strategies.

## Results

**Analysis and characterization of XO@1T-$MoS_2$/CFP catalysts.** The process of constructing XO@1T-$MoS_2$ on highly conductive carbon fiber paper (CFP) using Anderson-type POM nanoclusters ($XMo_6$) through a simple sulfuration hydrothermal reaction is illustrated in Fig. 1. The purity of $XMo_6$ precursors was confirmed by Fourier-transform infrared spectroscopy (FT-IR) analysis (Supplementary Fig. 1). Post-synthesis analysis via scanning electron microscopy (SEM) shows the CFP coated with ultrathin X@1T-$MoS_2$ nanosheet structures (Fig. 2a and Supplementary Fig. 2a–c). A more detailed nanosheet structure of X@1T-$MoS_2$ is observed via transmission electron microscopy (TEM) (Fig. 2b and Supplementary Fig. 3). Uniform elemental distribution of the FeO@1T-$MoS_2$, CoO@1T-$MoS_2$, and NiO@1T-$MoS_2$ nanosheets are confirmed via energy dispersive X-ray spectroscopy (EDX) analysis (Fig. 2c and Supplementary Fig. 4). Mo, S, O, Fe, Co, and Ni elemental content in the nanosheets are confirmed by inductively coupled plasma-atomic emission spectrometry (ICP-AES) analysis and are summarized in Supplementary Table 1. The experimental analysis determined the molar ratios of X:Mo in FeO@1T-$MoS_2$, CoO@1T-$MoS_2$, and NiO@1T-$MoS_2$ nanosheets to be 1:5.93, 1:5.94, and 1:5.96, respectively. These results are very similar to the elemental ratio of their $XMo_6$ precursors (X:Mo = 1:6). This indicates the chemical doping ratio is able to be well controlled using the suggested template for synthesis.

Crystalline structures of various XO@1T-$MoS_2$ nanosheets confirmed by X-ray diffraction (XRD) analysis include results shown in Supplementary Fig. 2d. NiO@1T-$MoS_2$ nanosheets show clear characteristic diffraction peaks at $2\theta = 13.60°$, 32.39°,

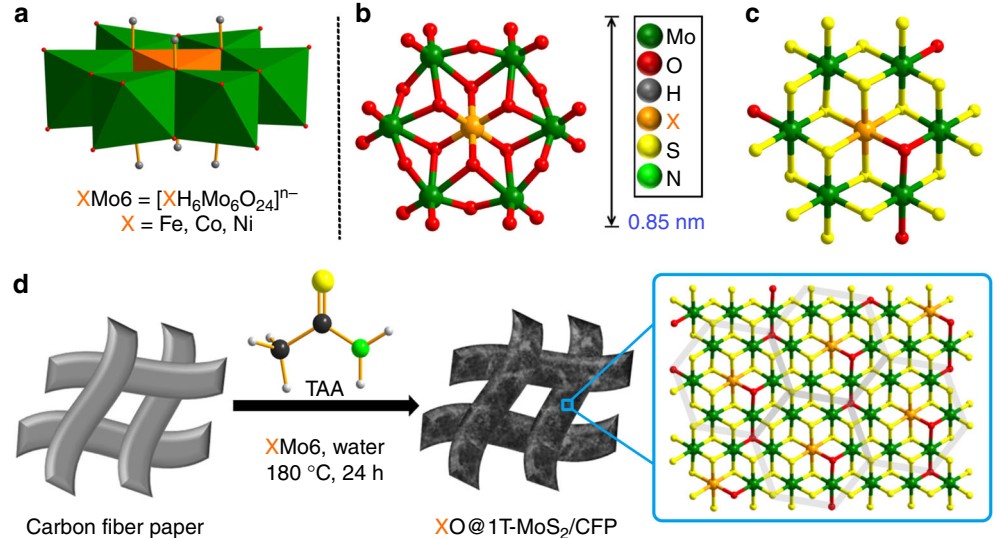

**Fig. 1** Structure of POM precursors and preparation of the XO@1T-MoS$_2$ nanosheets. **a** Polyhedral representation of the XMo$_6$ precursors. **b** Ball and stick representation of the XMo$_6$ precursors. **c** The mode structure of XO@1T-MoS$_2$. **d** Schematic illustration of the preparation of atomic scale transition metal and oxygen co-doped 1T-MoS$_2$ nanosheets on carbon fiber paper (CFP), XO@1T-MoS$_2$/CFP, by incomplete sulfuration of XMo$_6$ (green: Mo; yellow: S; orange: transition metal X; red: O)

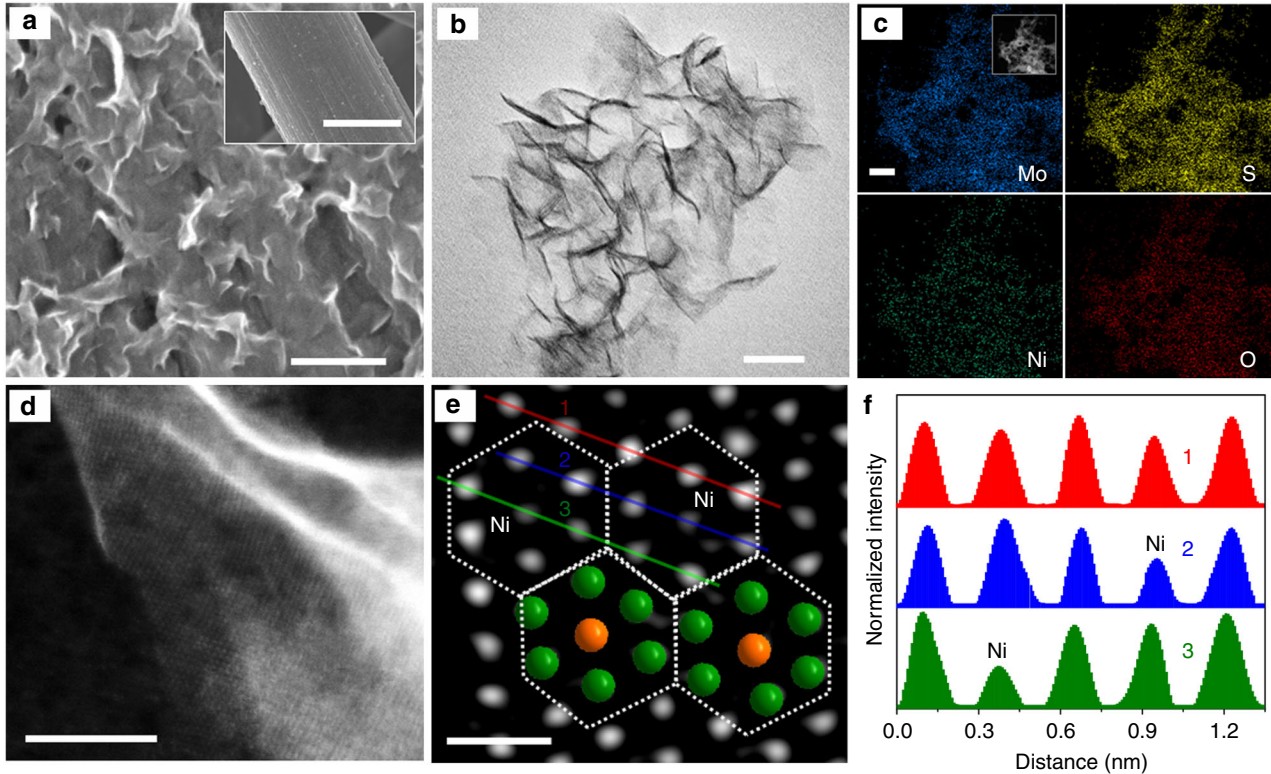

**Fig. 2** Structure characterizations of NiO@1T-MoS$_2$. **a** SEM image, scale bar: 200 nm (inset: low magnification SEM image, scale bar: 5 μm). **b** TEM image (scale bar: 20 nm). **c** EDX mappings (scale bar: 1 μm). **d** High-angle annular dark-field (HAADF) STEM image (scale bar: 5 nm). **e** Aberration-corrected atomic resolution HAADF-STEM image (scale bar: 0.5 nm). The white dotted hexagons show the NiMo$_6$ units in NiO@1T-MoS$_2$ (green: Mo; orange: Ni). **f** Intensity profiles along the lines indicated in image e

and 35.21° (see the pink curve in Supplementary Fig. 2d), which corresponds to the (002), (100), and (102) planes of hexagonal MoS$_2$ (PDF#75-1539). The peak at $2\theta = 13.60°$ suggests NiO@1T-MoS$_2$ has a stacked, multilayered structure, with $6.5 \pm 0.1$ Å spacing. This spacing is larger than the spacing seen in bulk 2H-MoS$_2$ (6.3 Å, PDF#75-1539). Similar XRD results of 1T-MoS$_2$ nanosheets (M-MoS$_2$) have been reported by Chen et al.[56].

TEM and high-angle annular dark-field scanning TEM (HAADF-STEM) images support the ultrathin nature of the Ni@1T-MoS$_2$ nanosheets by showing that 1T-MoS$_2$ is composed

of only a single to a few layers (see Fig. 2b, d, and Supplementary Fig. 5). These layers are composed of stacked 1T-MoS$_2$ layers as proven by diffractogram analysis and the inverse Fourier transformation analysis of images (Supplementary Fig. 5a–g). The lattice fringe spacing of NiO@1T-MoS$_2$ measured 2.60 nm for a five-layer nanosheet, indicating the interlayered spacing is 6.50 Å (Supplementary Fig. 5h), which is consistent with the value obtained with XRD. Aberration-corrected atomic resolution HAADF-STEM images used to investigate the atomic arrangement of NiO@1T-MoS$_2$ are shown in Fig. 2e. NiO@1T-MoS$_2$ exhibited hexagonal intensity variations, which is in agreement with the reported STEM image for 1T-MoS$_2$ single layer[57]. These intensity variations observed in the Ni centers and Mo atoms can be confidently assigned across various regions of the sample (Fig. 2f). Results show the hexagon units (white dotted hexagons in Fig. 2e) derived from the NiMo$_6$ precursors are well retained in NiO@1T-MoS$_2$ nanosheet (Fig. 2e, f).

Synchrotron-based X-ray absorption spectroscopy (XAS) was used to characterize the local structure of 1T-MoS$_2$ and XO@1T-MoS$_2$ analogs. Figure 3a compares the X-ray absorption near edge structure (XANES) spectra of 2H-MoS$_2$ to NiO@1T-MoS$_2$ at the Mo K-edge. Notable difference in the absorption edge corresponding to 1s–5p transitions was observed between these two samples, including the varying transition intensity and slight edge energy shift. This suggests a difference in local coorridnation enviroments surrounding the Mo center. Difference in local coordination geometry was further confirmed by their Fourier-transformed R-space spectra (Fig. 3b), where 2H-MoS$_2$ shows a prominent second-shell Mo–Mo scattering feature that is inconspicuous in NiO@1T-MoS$_2$. This difference may be attributed to the Mo in 1T-MoS$_2$ being octahedrally-coordinated to relatively large S scatterers, with respect to the trigonal-prismatic coordination in 2H-MoS$_2$ (see Supplementary Fig. 6c–d and Supplementary Table 2)[52]. On the other hand, both XANES and extended X-ray absorption fine structure (EXAFS) spectra of NiO@1T-MoS$_2$ resemble that of 1T-MoS$_2$, CoO@1T-MoS$_2$, and FeO@MoS$_2$ (Supplementary Fig. 6a–d), suggesting very similar local structure around the Mo center for XO@1T-MoS$_2$ analogs, consistent with XRD results. Quantitative FEFF fitting results further confirm the similar Mo local structure in all XO@1T-MoS$_2$ samples with similar interatomic distances within the uncertainty of the published crystal structures for 1T-MoS$_2$ (Supplementary Fig. 6c–d and Supplementary Table 2).

In addition to the Mo edge, the XAS spectrum of NiO@1T-MoS$_2$ at the Ni K-edge was measured to examine its local structure at Ni center. As shown in Fig. 3c, the XANES spectrum shows a sharp 1s–4p white line and very weak quadrupole-allowed pre-edge feature (inset of Fig. 3c), suggesting the octahedral coordination geometry of Ni[58]. Quantitative EXAFS fitting of the EXAFS spectrum (Fig. 3d and Supplementary Table 5) further supports Ni's similar local structure to Mo, with a fitted Mo–S coordination number of 4.13 in the Mo K-edge (Supplementary Table 2) and Ni–S coordination number of 4.27 (Supplementary Table 5) in the Ni K-edge. Moreover, a small contribution of Ni–Mo can be observed in the second-shell, implying proximity of Ni centers to Mo centers. Importantly, the formation of NiO may be excluded by comparing the XAS of the NiO reference with NiO@1T-MoS$_2$ (Supplementary Fig. 7). These results together suggest succesful incorporation of Ni and O into the host 1T-MoS$_2$ lattice without significant local structure disruptions or phase changes, in agreement with XRD.

A crucial difference between 1T-MoS$_2$ and 2H-MoS$_2$ lies within the symmetry of the S atoms in their structures. Changes in the symmetry of S atoms in these structures lead to significant differences in their characteristic Raman features. As shown in Fig. 4a, the 2H-MoS$_2$ (red curve) has three distinct vibrational modes located at 380, 406, and 450 cm$^{-1}$. These vibrational modes correspond to the $E^1_{2g}$, $A_{1g}$, and the longitudinal acoustic phonon modes, respectively[51]. The disappearance of the 2H-related peaks and the emergence of new characteristic Raman peaks at 147, 214, 236, 283, and 335 cm$^{-1}$ (Fig. 4a) associated with the phonon modes in 1T-MoS$_2$ confirms the formation of a pure 1T-MoS$_2$ nanosheet[50–52,59]. The zoomed-in spectra in the region of 100–250 cm$^{-1}$ and 250–350 cm$^{-1}$ do show small peak shifts and peak intensity changes upon the secondary X metal doping (Supplementary Fig. 20). We note that substituting Mo in 1T-MoS$_2$ by the secondary metal X with different chemical property and atomic mass (Fe, Co, Ni) will in principle lead to modification of vibration spectrum, as indeed indicated by our first-principles local vibration frequencies calculations. The moderate frequency shifts and the weak vibration intensity owing to the small amount of X may be responsible for the observed slight changes of Raman spectra of X@1T-MoS$_2$ (X = Fe, Co, Ni) (Fig. 4a and Supplementary Fig. 20). Considering these small peak changes, we may conclude that the introduction of secondary metals (Fe, Co, and Ni) has little influence on the pristine 1T-MoS$_2$ structures (Fig. 4a), consistent with the XAS and XRD results.

To evaluate the chemical composition and valence state of NiO@1T-MoS$_2$/CFP, the NiO@1T-MoS$_2$ nanosheet layers were further studied with X-ray photoelectron spectroscopy (XPS). As shown in Supplementary Fig. 8a, the XPS spectrum of NiO@1T-MoS$_2$ indicates the presence of Mo, S, O, and Ni. The high-resolution Mo 3d spectra of the NiO@1T-MoS$_2$ nanosheets was mainly deconvoluted into two peaks (Fig. 4b). The two characteristic peaks of Mo 3d$_{5/2}$ (229.22 eV) and Mo 3d$_{3/2}$ (232.41 eV) suggest the dominance of Mo$^{4+}$ in NiO@1T-MoS$_2$[49,56]. Moreover, the positive shift in the binding energies (~0.5 eV) of Mo 3d$_{5/2}$ and Mo 3d$_{3/2}$ in NiO@1T-MoS$_2$, with respect to the corresponding peaks in 1T-MoS$_2$ (Supplementary Fig. 8b), indicate doping Ni influences Mo's electronic structure in 1T-MoS$_2$. The weak peak located at 235.94 eV corresponds to the production of Mo$^{6+}$. This is possibly due to the partial oxidation of Mo$^{4+}$ on the catalyst's surface[47,60,61].

The S 2p spectrum (Fig. 4c) was deconstructed into four peaks assigned to S$^{2-}$ [162.0 eV (S 2p$_{3/2}$) and 163.2 eV (S 2p$_{1/2}$)], S$_2^{2-}$ (164.5 eV), and S$^{6+}$ (169.5 eV)[49,56,62]. The S$^{6+}$ (169.5 eV) may be attributed to a physically-absorbed sulfate, derived from the sulfate moieties on the surface of NiO@1T-MoS$_2$. The absorbed sulfate may be easily removed by washing with DI water, with washed catalysts exhibiting similar catalytic activity as unwashed catalysts (Supplementary Fig. 21). The S 2p peaks also exhibit a positive shift of ~0.5 eV (S 2p$_{3/2}$) and 0.2 eV (S 2p$_{1/2}$), relative to pristine 1T-MoS$_2$ nanosheets (Supplementary Fig. 8c). As shown in Fig. 4d, the high-resolution XPS spectrum of Ni 2p exhibits two main peaks located at 857.8 and 875.9 eV, which belong to Ni 2p$_{3/2}$ and Ni 2p$_{1/2}$, respectively. These peaks suggest the formation of Ni–S bonds[63,64]. The peak located at 854.6 eV belongs to the Ni–O bonds[65], indicating intercalation of oxygen between the Ni and Mo atoms (Ni–O–Mo). Furthermore, according to the peaks shifts shown in Supplementary Fig. 8, results indicate doping Ni and Co atoms provides a stronger influence on the electronic structure of MoS$_2$ than doping Fe atoms does. Detailed comparisons are provided below Supplementary Fig. 8. Once again, all results discussed herein indicate the facile POM template strategy atomically engineers transition metal active sites into ultrathin 1T-MoS$_2$ nanosheets successfully.

**Evaluation of electrochemical HER activities**. Experimental results show a noticeable enhancement in HER activity for 1T-MoS$_2$ samples doped with oxygen and Fe, Co, and Ni (see Fig. 5a).

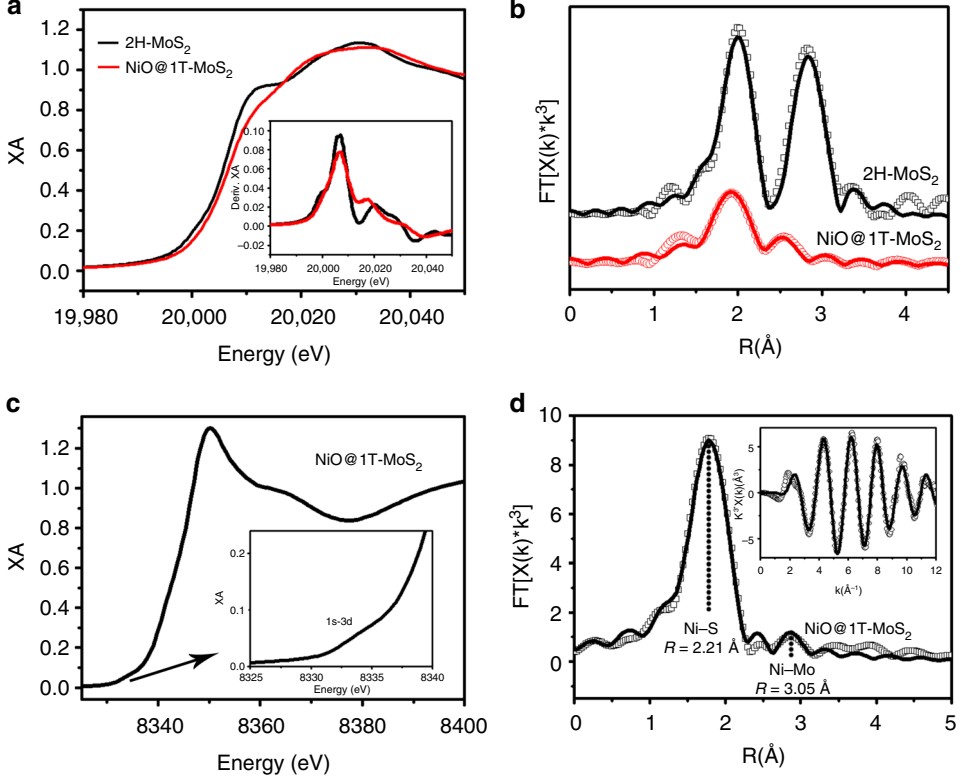

**Fig. 3** X-ray absorption analysis. **a** The normalized Mo K-edge XANES spectra and first derivative inset. **b** Comparison of R-space data and best fit lines. **c** Ni K-edge XANES spectrum for NiO@1T-MoS₂ with enlarged pre-edge region inset. **d** Corresponding EXAFS spectrum fitting of NiO@1T-MoS₂ in R-space with K-space inset

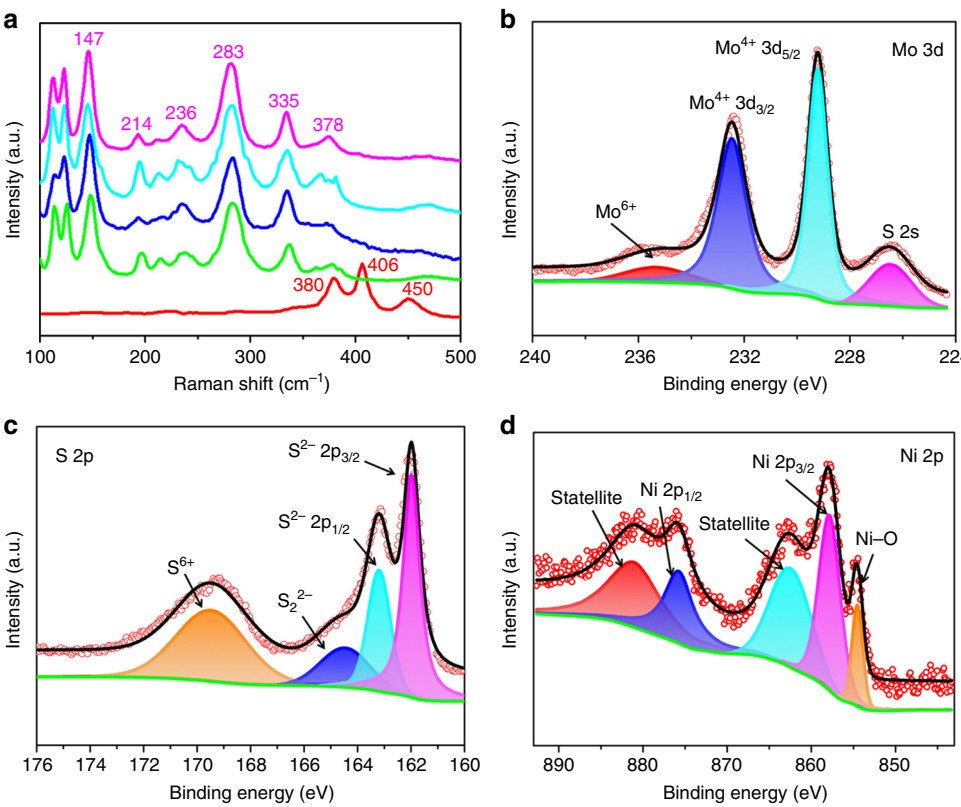

**Fig. 4** Raman and XPS spectra. **a** Raman spectra of each XO@1T-MoS₂ nanosheet (red: 2H-MoS₂; green: 1T-MoS₂; blue: FeO@1T-MoS₂; light blue: CoO@1T-MoS₂; pink: NiO@1T-MoS₂). **b–d** High-resolution XPS signals of **b** Mo 3d; **c** S 2p; **d** Ni 2p for the NiO@1T-MoS₂ nanosheet

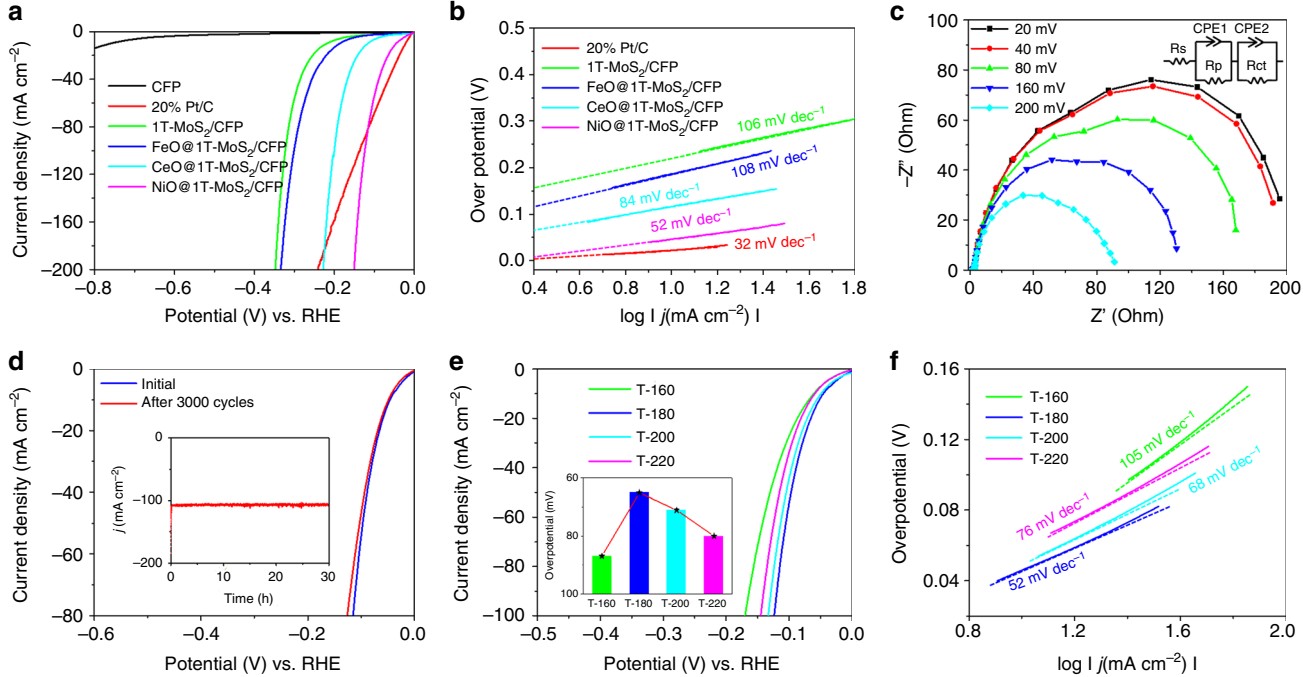

**Fig. 5** HER catalytic performances and EIS responses for NiO@1T-MoS$_2$. **a** Polarization curve of all catalysts in 1.0 M KOH (scan rate: 5 mV s$^{-1}$ under a three-electrode configuration). **b** Tafel plots. **c** Electrochemical impedance spectroscopy (EIS) of NiO@1T-MoS$_2$ at various overpotentials. **d** Stability tests of NiO@1T-MoS$_2$ (inset: the long-term durability tests at $\eta = 130$ mV for 30 h). **e** Polarization curves (inset: overpotentials to drive 20 mA cm$^{-2}$). **f** The corresponding Tafel slopes of NiO@1T-MoS$_2$ synthesized at various temperatures: 160 °C (T-160), 180 °C (T-180), 200 °C (T-200), and 220 °C (T-220) samples

As the control, the 1T-MoS$_2$ catalyst demonstrates a 133 mV onset potential and 10 mA cm$^{-2}$ cathodic current density at 219 mV overpotential ($\eta_{10}$) (Supplementary Table 4). Of the samples studied, NiO@1T-MoS$_2$ exhibited the highest HER activity. NiO@1T-MoS$_2$ required a minor overpotential ($\eta_{10} = 46$ mV) to obtain a current density of 10 mA cm$^{-2}$. In comparison, the overpotentials for the other samples studied were much larger (1T-MoS$_2$ = 219 mV, FeO@1T-MoS$_2$ = 187 mV, CoO@1T-MoS$_2$ = 117 mV). NiO@1T-MoS$_2$ even produced results superior to 20% Pt/C at $\eta_{10}$ > 118 mV. The data collected in this experiment was compared with applicable results obtained in literature, including: Ni–MoS$_2$ (98 mV)[48], T-MoS$_2$ (~300 mV)[21], MoS$_2$/Ni$_3$S$_2$ (110 mV)[46], Ni–Co–MoS$_2$ (155 mV)[47], with the full list available in Supplementary Table 7. Notably, Hou et al. recently converted the same Anderson-type polyoxometalate into oxygenated CoS$_2$–MoS$_2$ heteronanosheet but instead used thiourea for sulfuration at 200 °C and further thermal treatment at 400 °C, in which CoS$_2$ and MoS$_2$ form completely separated phases instead of atomically doping 1T-MoS$_2$ with secondary metals. Moreover, a much higher overpotential (97 mV) was achieved by the best performed CoMoS catalyst[26]. Comparison of all results shown in Supplementary Table 7 show NiO@1T-MoS$_2$ exhibits the best performance for HER. Most catalysts reported for comparison were prepared using one-pot reaction methods. These methods either suffered from dopant formation limited to edge sites (2H-MoS$_2$)[24], or from insertion of uncontrolled amounts of dopants into unclear dopant locations[22,25,26]. Thus, our method shows great potential for preparing a low-cost and highly active HER catalyst.

HER kinetics were further evaluated using Tafel plots (Fig. 5b). NiO@1T-MoS$_2$ shows a Tafel slope slightly higher than 20% Pt/C, but much lower than the other catalysts tested in this experiment. The electrocatalysts' exchange current densities ($j_0$) were calculated to evaluate the inherent HER activities (see Supplementary Table 4). Results indicate the exchange current density of NiO@1T-MoS$_2$ (0.44 mA cm$^{-2}$) is very close to 20% Pt/C (0.69

mA cm$^{-2}$). Electrochemical impedance spectroscopy (EIS) analyzed at various overpotentials show similar impedance properties at each overpotential. This suggests that similar electrochemical processes proceed in 1.0 M KOH at all overpotentials tested (Fig. 5c). EIS for 1T-MoS$_2$ and XO@1T-MoS$_2$ performed at an overpotential of 200 mV are provided as well in Supplementary Fig. 9. All results are fitted by a simplified equivalent circuit (inset in Fig. 5c). NiO@1T-MoS$_2$ shows a much lower charge-transfer resistance ($R_{ct}$) value in comparison with all other catalysts tested (Supplementary Table 4), suggesting improved charge-transfer properties and HER kinetics in NiO@1T-MoS$_2$. To confirm the HER activity is driven by the catalytic sites on NiO@1T-MoS$_2$ and not the supportive material, bare CFP was tested as well, with results proving negligible HER activity in the CFP composites (Fig. 5a).

The electrochemical double-layer capacitances ($C_{dl}$) measured via cyclic voltammetry (CV) (Supplementary Fig. 10) were employed to evaluate the active surface areas[50,56]. Since $C_{dl}$ is proportional to the electrochemical active surface area (ECSA), it may be used to identify different electrocatalytic active sites[66]. The $C_{dl}$ value of 1T-MoS$_2$ is ~1.09 mF cm$^{-2}$ (Supplementary Table 4), while FeO@1T-MoS$_2$, CoO@1T-MoS$_2$, and NiO@1T-MoS$_2$ all show higher $C_{dl}$ values (11.38, 16.37, and 18.32 mF cm$^{-2}$, respectively). In summary, NiO@1T-MoS$_2$ presents the best HER activity compared to bare 1T-MoS$_2$ and XO@1T-MoS$_2$ (X = Fe, Co). These results indicate doping Ni into 1T-MoS$_2$ provides more HER active sites (Fig. 5) and account for its increase in $C_{dl}$ values (Supplementary Fig. 10). Furthermore, NiO@1T-MoS$_2$ shows lower resistance (Supplementary Fig. 9 and Supplementary Table 4). This most likely stems from the increase in active sites caused by precisely doping Ni and O into the MoS$_2$ nanosheets, as well as improvement in charge transport properties exhibited by MoS$_2$ in the 1T-phase.

Long-term stability of NiO@1T-MoS$_2$ by cycling NiO@1T-MoS$_2$ continuously for 3000 cycles (from 0 to −0.2 V vs RHE, scan rate: 100 mV s$^{-1}$, 1.0 M KOH) shows negligible changes in

the polarization curve (Fig. 5d). Meanwhile, chronoamperometry (CA) analysis in alkaline conditions shows a stable HER current versus time plot over a 30 h period (~107 mA cm$^{-2}$ at 130 mV overpotential, shown in the inset of Fig. 5d). After 30 h, even at a low overpotential of −130 mV, a turnover number (TON) of 75,600 and turnover frequency (TOF) of 0.70 s$^{-1}$ are both obtained via analysis of the catalyst loading using inductively coupled plasma-atomic emission spectroscopy (ICP-AES). TOF and TON calculation details may be found in the Supplementary Methods sections. Applicable SEM and TEM images, as well as the Raman spectrum, reveal the ability of NiO@1T-MoS$_2$ to retain its morphology and structural integrity after long-term HER stability testing in 1.0 M KOH (Supplementary Fig. 11). Analysis of the Faradic efficiency shows the amount of H$_2$ generated by NiO@1T-MoS$_2$ to be consistent with its theoretical value, with an average Faradic efficiency around 99.5% (Supplementary Fig. 12).

**Structure-activity relationship**. To uncover the structure-activity relationship in NiO@1T-MoS$_2$, a series of factors that have a possibility of influencing the HER activity were investigated, such as the support substrates, loading mass, and reaction temperature (Supplementary Figs. 13–15). To study the influence of support substrates on HER activity, the bare CFP, NiO@1T-MoS$_2$/CFP, and NiO@1T-MoS$_2$ catalysts on a glassy carbon electrode (NiO@1T-MoS$_2$/GCE) were compared (Supplementary Fig. 13a). Results demonstrate replacing CFP with GCE results in lower HER activity. Therefore, in situ growth of NiO@1T-MoS$_2$ nanosheets onto CFP is essential to optimize HER performance. An analysis of different loading masses of NiO@1T-MoS$_2$ on CFP (Supplementary Fig. 13b) determine the optimal loading mass to be 1.02 mg cm$^{-2}$. It is postulated that the electrocatalysts tend to aggregate when the loading mass is in excess and tend to be less active when the loading mass is too low (see Supplementary Fig. 14).

As the sulfuration process becomes less sufficient with decreasing reaction temperature, an increase amount of oxygen incorporated into the original Ni–O or Mo–O bonds occurs. Thus, changing the temperature used during synthesis clearly impacts catalytic activity and HER performance. As shown in Supplementary Figs. 15, the four samples studied (T-160, T-180, T-200, and T-220) all exhibit similar nanosheet structures. Raman spectra of these samples to investigate their structural information is provided in Supplementary Fig. 16a. As the synthetic temperature starting at 200 °C increases, a weak characteristic Raman peak at 406 cm$^{-1}$ (gray area in Supplementary Fig. 16a) associated with the A$_{1g}$ first-order phonon modes of 2H phase emerges. Further assessment of the temperature-varied samples in 1.0 M KOH electrolyte conducted showed that the T-180 sample exhibits the lowest overpotential (65 mV for 20 mA cm$^{-2}$ current density) of all the samples tested. Results are shown in Fig. 5e. This result suggests a volcano-like relationship between overpotential and the oxygen incorporation content (inset in Fig. 5e). The corresponding Tafel plots (Fig. 5f) indicate that the water dissociation step kinetics are effectively facilitated for T-180 (52 mV dec$^{-1}$). The inferior HER performances of T-200 and T-220 are most likely due to the emergence of 2H-MoS$_2$ phase witnessed in Raman analysis of the samples at higher temperatures (Supplementary Fig. 16a). T-160's lower HER activity is likely due to the sulfuration process being less efficient at 160 °C, which is demonstrated by the higher oxygen content present upon EDX analysis (Supplementary Fig. 16b) and elemental analysis comparison with T-180 (Supplementary Table 1). The exact content of oxygen atoms in the samples obtained via elemental analysis are summarized in Supplementary Table 1.

Results reveal that the oxygen content varies from 6.38 to 1.35% with an increase in temperature. It should be noted that all four samples were exposed to air at similar conditions before elemental analysis, thus suggesting possible oxygen incorporation (incomplete sulfuration) in Ni@1T-MoS$_2$ outside of surface oxidation by air.

Since XPS results indicate the presence of Ni–O–Mo bonding, which may be critical electrocatalytic moieties, we turned to XAS measurements to explore the effects of synthesis temperature on the local structure of Ni (Supplementary Fig. 17). Notably, a trend is observed in the first-shell scattering peak in R-space, where higher synthesis temperature leads to more intense scattering and an increase in distance. The quantitatively fit parameters in Supplementary Table 5 show that the effective number of Ni–S bonds increase with synthesis temperature, causing the witnessed R-space intensity trend. This result is in direct agreement with ICP-AES results, where lower temperature results in more O incorporation (and, therefore, less sulfurization). Although there is no clear trend in the Ni–Mo effective coordination number with synthesis temperature due to weak second-shell scattering, the observation of Ni–Mo single scattering in all samples suggests the presence of Ni–O–Mo moieties.

The electrocatalytic performances of NiO@1T-MoS$_2$ for HER in acidic conditions (0.5 M H$_2$SO$_4$) is shown in Supplementary Fig. 18. NiO@1T-MoS$_2$ exhibits an enhanced HER performance compared to 1T-MoS$_2$. The durability test (Supplementary Fig. 18c) shows no obvious shift in the polarization curve after 3000 CV cycles. The stability of NiO@1T-MoS$_2$ is further supported by a stable hydrogen evolution current versus time plot over 30 h (~90 mA cm$^{-2}$ at 210 mV overpotential, inset of Supplementary Fig. 18c), indicating its prolonged stability under acidic conditions as well. Moreover, the onset potential and the Tafel slope of the NiO@1T-MoS$_2$ are very similar to the undoped 1T-MoS$_2$. The above results imply that the HER performance of NiO@1T-MoS$_2$/CFP in acidic electrolyte is inferior to its performance in alkaline electrolyte. Interestingly, the oxygen incorporations in NiO@1T-MoS$_2$ nanosheets greatly improves the HER performance in alkaline electrolyte. Nevertheless, the HER performance in acidic electrolyte is suppressed.

**First-principles calculations**. Further insight into the underlying mechanism of NiO@1T-MoS$_2$ in HER was deduced by first-principles density functional theory (DFT) calculations. In principle, the reaction pathway for a HER process in alkaline media involves four steps: the initial catalyst-H$_2$O contact step, the H$_2$O dissociation (or Volmer) step, the formation of H* intermediates step, and the H$_2$ generation (Tafel or Heyrovsky) step. The free energy differences between the first and second steps, $\Delta G(H_2O)$, and between the third and fourth steps, $\Delta G(H*)$ are widely considered useful for understanding HER catalytic activity[46,48]. The free energy diagrams were calculated on the surface of the 1T-MoS$_2$ catalysts with different transition metals X doped into each catalyst. We constructed the X-doped 1T-MoS$_2$ structure with the stoichiometry of XMo$_6$S$_{14}$ based on the HAADF-STEM image of Fig. 2e for simulation. The structure is composed of the hexagonal XMo$_6$S$_{14}$ motifs under the crystalline symmetry of 1T-MoS$_2$, where each X atom is surrounded by six equivalent Mo atoms and each Mo atom is surrounded by one X and five Mo atoms (Fig. 6a).

The calculated Gibbs free energy diagrams (with the explicit data in Supplementary Table 6) are given in Fig. 6b. Results show a decrease in $\Delta G(H*)$ after doping 1T-MoS$_2$ with transition metals, i.e., 0.291 eV for Fe@1T-MoS$_2$, 0.247 eV for Co@1T-MoS$_2$, and 0.158 eV for Ni@1T-MoS$_2$. All of the results are lower than 0.698 eV seen in pristine 1T-MoS$_2$. Particularly,

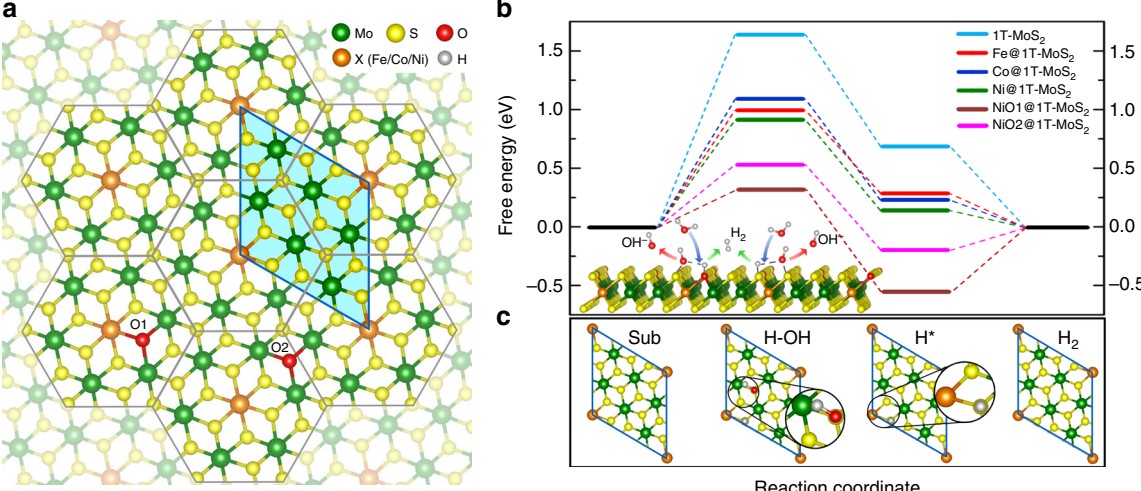

**Fig. 6** First-principles calculations of the doping effect on HER performance. **a** The monolayer structure of XO@1T-MoS$_2$ formed by the XMo$_6$ hexagon motifs codoped by O. **b** Free energy diagrams on the surface of different catalysts in alkaline solution, with the smallest repeating unit cell used in calculations indicated in **a**. **c** The structures of predicted intermediates that bind to water and hydrogen. The green, yellow, orange and red spheres represent Mo, S, X (Fe, Co, Ni) and O atoms, respectively

the $\Delta G(H^\star)$ of Ni@1T-MoS$_2$ is close to 0 eV, implying its superior ability to promote H$^\star$ desorption. Turning to the energy barrier of H$_2$O dissociation $\Delta G(H_2O)$, the pristine 1T-MoS$_2$ demonstrated an extremely high value of 1.650 eV, which significantly hinders the dissociation of H$_2$O to H$^\star$ intermediates and leads to sluggish HER kinetics. With the transition metals doped into the catalysts, $\Delta G(H_2O)$ decreases to 1.005 eV for Fe@1T-MoS$_2$, 1.098 eV for Co@1T-MoS$_2$, and minimizes at 0.925 eV for Ni@1T-MoS$_2$. This tendency is in agreement with the above experimental results.

Furthermore, controlling the synthetic temperature allows the Ni@1T-MoS$_2$ samples to exhibit different oxygen incorporations to produce NiO@1T-MoS$_2$ (see Supplementary Fig. 16b). Investigation of the cooperative effect of oxygen incorporation into Ni@1T-MoS$_2$ results in $\Delta G(H_2O)$ further reduced to 0.538 eV. Meanwhile, $\Delta G(H^\star)$ decreases to a negative value with a magnitude of 0.186 eV. This is comparable to previous theoretical investigations of the platinum (Pt) (111) surface for alkaline HER[28]. The reaction's schematic representation is shown in Fig. 6c, along with a more detailed hydrogen generation process given in the bottom of Fig. 6b. By carefully examining many possible HER reaction pathways on the catalyst surfaces, the energetically favored ones were identified, as shown in Supplementary Fig. 19. In addition to the most stable pathway, we find another metastable pathway of NiO@1T-MoS$_2$ with slightly higher energy was able to reduce $\Delta G(H_2O)$ to 0.324 eV (Supplementary Table 6). These results further suggest the Ni–O–Mo motif is the most possible activation site for the water dissociation step. However, this suppresses the H$^\star$ desorption process. This may explain why the T-160 sample with high oxygen doping shows lower HER activity than the T-180 with suitable oxygen incorporation. As shown in Fig. 6b, oxygen incorporations increase the binding of hydrogen on the surface of NiO@1T-MoS$_2$, thus suppressing HER kinetics in an acidic electrolyte (Supplementary Fig. 18). Inversely, for 1T-MoS$_2$, the hydrogen binding is too weak for proceeding the HER due to the high free energy for proton reduction. In this case, the oxygen incorporations into 1T-MoS$_2$ improve the HER kinetics in acidic electrolyte. As the result, the integrated incorporations of Ni and O atoms make the HER performance of NiO@1T-MoS$_2$ comparable to that of the O incorporated 1T-MoS$_2$ in acidic electrolyte (Supplementary Fig. 18). In general, these results indicate that the synergy of uniform Ni doping and suitable oxygen incorporations in NiO@1T-MoS$_2$ nanosheets not only improve the sluggish alkaline HER kinetics, but also provides abundant HER active sites.

The high HER activity and stability of NiO@1T-MoS$_2$/CFP may be attributed to its good conductivity and electrical transport efficiency to facilitate HER kinetics. Additionally, the chemical structure regulation and stabilization effect of intercalating transition metals and suitable oxygen into 1T-MoS$_2$ through precise engineering of atomic activation sites contributes to the generation of more active sites for water dissociation and thus facilitate electrode kinetics. Moreover, the direct growth of NiO@1T-MoS$_2$ with suitable loading mass on conductive CFP substrates leads to fast interdomain electron transport.

**Discussion**

In summary, atomically engineering activation sites on 1T-MoS$_2$ nanosheets has been achieved, providing an opportunity to significantly modulate the electronic structure and accelerate the sluggish kinetics in HER. In this work, a series of transition metal and oxygen co-doped 1T-phase MoS$_2$ nanosheets (denoted as XO@1T-MoS$_2$) are obtained by employing various Anderson-type POM nanoclusters as precursors. NiO@1T-MoS$_2$/CFP (T-180), demonstrated the most outstanding HER activity of all the various catalysts tested herein, with a low onset overpotential (~0 mV), a small Tafel slope (52 mV dec$^{-1}$), a high exchange current density (0.44 mA cm$^{-2}$), and good stability (over 30 h) under strong alkaline conditions. In a comparison of HER performance, the NiO@1T-MoS$_2$/CFP (T-180) catalyst's results outshined the state-of-the-art 20% Pt/C catalyst used commercially today, when the overpotential is more negative than −118 mV. The combined experimental and theoretical results show that precisely co-doping Ni and O atoms into ultrathin 1T-MoS$_2$ nanosheets effectively lowers the kinetic energy barrier required for the initial water dissociation step, meanwhile facilitating the process of hydrogen generation from their intermediate states. The ability to design electrocatalysts with atomic level electronic modulations may be challenging, but it is a vital to advancing our understanding of single-atom catalytic activation sites at the experimental level to help progress the future of designing optimal catalytic systems.

## Methods

**Materials and characterization**. All chemicals were obtained as reagent grade chemicals from Adamas-beta® unless noted. IR spectrum was measured by using KBr pellets and recorded on a Perkin Elmer FT-IR spectrometer. Elemental analyses of H and N were performed on PERKIN ELMER CE-440. Elemental analyses of Mo, S, Fe, Co, and Ni were performed by ICP-AES on Thermo IRIS Intrepid II. 1T-$MoS_2$ and XO@1T-$MoS_2$ samples without CFP were digested by heating digestion method before ICP-AES tests as the following: take NiO@1T-$MoS_2$ for example, 40 mg NiO@1T-$MoS_2$ was dissolved by 6 mL $HNO_3$ (65 wt%) and 3 mL HF (40 wt%) in a Teflon container and then put into an electric heater at 160 °C. The as-obtained solution was then transferred and diluted into a 100 mL volumetric flask with ultra-pure water for ICP-AES testing. Powder X-ray diffraction characterization was performed on a Bruker D8 Advance X-ray diffractometer using Cu-Ka radiation ($\lambda = 1.5418$ Å). The morphology and size of the nanostructured materials were characterized by a HITACHI H-7700 TEM with an accelerating voltage of 100 kV, and a FEI Tecnai G2 F20 S-Twin high-resolution (HR) TEM, operating at 200 kV on a HITACHI S-5500. Scanning electron microscopy (SEM) with energy dispersive X-ray spectroscopy (EDX) equipment was conducted on a LEO 1530. High-resolution high-angle annular dark-field scanning TEM (HAADF-STEM) images were recorded by a double-corrected JEOL Grand ARM-300CF (60–300 keV), equipped with a cold field emission electron source, operated at 80 keV. The high-angle annular HAADF-STEM images were processed by the Bandpass filter in the DigitalMicrograph software. X-ray photoelectron spectroscopy (XPS) experiments were carried out on a scanning X-ray microprobe (Quantera SXM, ULVAC-PHI. INC) operated at 250 kV and 55 eV with monochromated Al Kα radiation. The XPS spectra were calibrated with C1s = 284.8 eV and fitted using XPSPEAK41 software with Shirley background type and free parameters. Raman spectra were recorded using a HORIBA JY HR800 confocal Raman microscope employing an Ar-ion laser operating at 532 nm.

**Synthesis of FeMo₆ precursor**. The $(NH_4)_3[FeH_6Mo_6O_{24}]\cdot 7H_2O$ (FeMo₆) precursor was prepared according to a modified published procedure[54]. $(NH_4)_6Mo_7O_{24}\cdot 4H_2O$ (denoted $Mo_7$, 5.19 g, 4.2 mmol) was dissolved in water (80 mL) and then heated to 100 °C. $Fe(NO_3)_3\cdot 9H_2O$ (1.41 g, 3.5 mmol) was dissolved in water (20 mL), which was slowly added in the above solution with stirring. The pH of the mixing solution was maintained between 2.5 and 3. The mixture was kept heating and stirring to yield a deep brown solution. After 2 h, the crude product (3.85 g) was isolated via filtration. The yellowish targeted product (3.2 g, 54.3% yield based on Mo) was obtained by recrystallization in hot water (80 °C) two times, then dried at room temperature. Elemental analysis calcd (%) for $H_{32}N_3O_{31}FeMo_6$ (M = 1201.74 g $mol^{-1}$): H, 2.68; N, 3.50; Mo, 47.90; Found: H, 2.65; N, 3.48; Mo, 47.93. IR (KBr pellet, major absorbances, $cm^{-1}$): 3165 ($v_{as}NH$, m), 1640 (δOH, m), 1400 (δNH, s), 946 ($vMo=O$, vs), 884 ($vMo=O$, vs), 649 ($vMo$-O-Mo, vs), 572 ($vFe$-O-Mo, w).

**Synthesis of CoMo₆ precursor**. The $(NH_4)_3[CoH_6Mo_6O_{24}]\cdot 7H_2O$ (CoMo₆) precursor was prepared according to a modified published procedure[54]. $Mo_7$ (5.19 g, 4.2 mmol) was dissolved in water (80 mL) and then heated to 100 °C. Co $(SO_4)_2\cdot 7H_2O$ (1.13 g, 4 mmol) and 30% $H_2O_2$ (1 mL) were dissolved in water (20 mL), which was added to the above solution with stirring. The mixture was kept under heat and stirring to give rise to a deep green solution. The crude product (3.5 g) was isolated with evaporation and filtration. The green targeted product (2.8 g, 47.4% yield based on Mo) was obtained by recrystallization in hot water (80 °C) two times, then dried under vacuum. Elemental analysis calcd (%) for $H_{32}N_3O_{31}CoMo_6$ (M = 1204.83 g $mol^{-1}$): H, 2.68; N, 3.49; Mo, 47.78; Found: H, 2.62; N, 3.43; Mo, 47.75. IR (KBr pellet, major absorbances, $cm^{-1}$): 3182 ($v_{as}NH$, m), 1637 (δOH, m), 1403 (δNH, s), 943 ($vMo=O$, vs), 888 ($vMo=O$, vs), 650 ($vMo$-O-Mo, vs), 581 ($vCo$-O-Mo, w).

**Synthesis of NiMo₆ precursor**. The $(NH_4)_4[NiH_6Mo_6O_{24}]\cdot 5H_2O$ (NiMo₆) precursor was prepared according to a modified published procedure[54]. $Mo_7$ (5.19 g, 4.2 mmol) was dissolved in water (80 mL) and then heated to 100°C. $Ni(NO_3)_2\cdot 6H_2O$ (1.16 g, 4 mmol) was dissolved in water (20 mL), which was added to the above solution with stirring. The mixture was kept heating and stirring to give rise to a deep green solution. The crude product (5.4 g) was isolated with evaporation and filteration. The green targeted product (4.6 g, 79.1% yield based on Mo) was obtained by recrystallization in hot water (80 °C) two times, then dried under vacuum. Elemental analysis calcd (%) for $H_{32}N_4O_{29}NiMo_6$ (M = 1186.60 g $mol^{-1}$): H, 2.72; N, 4.72; Mo, 48.51; Found: H, 2.70; N, 4.66; Mo, 48.62. IR (KBr pellet, major absorbances, $cm^{-1}$): 3402 ($v_{as}OH$, m), 3152 ($v_{as}NH$, m), 1627 (δOH, m), 1402 (δNH, s), 929 ($vMo=O$, vs), 876 ($vMo=O$, vs), 635 ($vMo$-O-Mo, vs), 577 ($vNi$-O-Mo, w).

**Preparation of XO@1T-$MoS_2$/CFP**. Take the NiO@1T-$MoS_2$/CFP for example. The as-prepared NiMo₆ (50 mg, 0.042 mmol) precursors, thioacetamide (TAA, 80 mg, 1.065 mmol) and CFP (1 × 2 $cm^2$) were mixed in 10 mL $H_2O$, transferred into a 20 mL Teflon autoclave, and heated at 180°C for 24 h to give rise to the corresponding NiO@1T-$MoS_2$/CFP electrocatalyst. The loading amount of Ni–$MoS_2$ on CFP is about 1 mg $cm^{-2}$. The 1T-$MoS_2$/CFP, FeO@1T-$MoS_2$/CFP, and CoO@1T-$MoS_2$/CFP electrocatalysts were prepared according to the same protocol as NiO@1T-$MoS_2$/CFP, except for using $Mo_7$, FeMo₆, and CoMo₆ (0.042 mmol) precursors to replace NiMo₆

precursor, respectively. The NiO@1T-$MoS_2$/CFP catalysts with different oxygen incorporations were obtained by controlling the synthesis temperature. In this work, various synthesis temperatures (160, 180, 200, and 220 °C) were used to control the sulfuration process of NiO@1T-$MoS_2$/CFP, forming T-160, T-180, T-200 and T-220, respectively. The loading amount of these catalysts on CFP is also about 1 mg $cm^{-2}$ and was controlled by ultrasonic treatment. For comparison, the electrode containing 1 mg $cm^{-2}$ Pt/C on CFP was prepared as the following: 5 mg of 20% Pt/C powder was dispersed in 1 mL of solution containing 0.95 mL of ethanol and 50 µL of 0.5 wt% Nafion. Then the mixture was ultrasonicated for 30 min to generate a homogeneous slurry. Finally, an appropriate amount of Pt/C slurry was daubed uniformly over 1 $cm^2$ of area on a piece of 1 cm × 2 cm CFP and dried at room temperature for 24 h.

**X-ray absorption measurements and data fitting**. The Mo K-edge X-ray absorption spectra were collected at beamline 4-1 from Stanford Synchrotron Radiation Lightsource (SSRL). The X-ray fluorescence was detected by a Lytle-type fluorescence-yield ion chamber detector. In order to reduce background from elastic scattering, the Soller slits were aligned and fitted with suitable Z-1 filters. We ran the Mo K-edge extend X-ray absorption fine structure (EXAFS) in the range 19.778–20.887 keV in fluorescence mode with a step-size of 0.25 eV at the near edge. The Ni K-edge EXAFS was run in the range of 8.125–9.225 keV in fluorescence mode with a step-size of 0.25 eV at the near edge. All samples were prepared by placing a small amount of homogenized powder (via agate mortar and pestle) on 3M Kapton Polyimide tape, which was purchased from 3M (https://www.3m.com/).

The structural parameters around Mo and Ni atoms of XO@1T-$MoS_2$ were obtained by the least-squares curve parameter method with the ARTEMIS module of both IFEFFIT and USTCXAFS software packages[67]. The obtained parameters are summarized in Supplementary Tables 2, 3, and 5. The fitted R values are all within the uncertainty of the published crystallographic values[68,69]. FEFF fitting of the NiO reference spectrum was performed at R = 4, requiring second and third-shell single and multiple-scattering paths. The first-shell Ni–O and second-shell Ni–Ni distances both fit to values within the uncertainty of the published crystal structure that was used for the model[70]. The structural parameters around Ni atoms of NiO@1T-$MoS_2$ as a function of synthesis temperature were fit using the same model utilized at Mo K-edge, by replacing Mo with Ni.

**Electrochemical measurements**. The electrocatalytic HER activities of the prepared XO@1T-$MoS_2$/CFP and 1T-$MoS_2$/CFP catalysts were measured using a three-electrode configuration in $N_2$-saturated 1.0 M KOH (or 0.5 M $H_2SO_4$) aqueous solution (scan rate of 5 mV $s^{-1}$). The 20% Pt/C (Johnson Matthey) on CFP and bare CFP were measured as a comparison here. A saturated calomel electrode (SCE) and a graphite rod were applied as the reference and counter electrodes, respectively. All electrochemical tests are performed at room temperature. The HER electrochemical activities were conducted with a standard three-electrode system on a CHI660E potentiostat (CH Instruments, China). All the potentials in this work were referenced to a reversible hydrogen electrode (RHE) according to E (RHE) = E (SCE) + 0.2415 + 0.059 pH. Before the electrochemical tests, the fresh working electrode was cycled 50 times to stabilize the current. Linear sweep voltammetries (LSV) were tested in 1 M KOH at a scan rate of 5 mV $s^{-1}$. 95% iR compensation was applied for all linear sweep voltammetry (LSV) measurements, unless noted otherwise. Additionally, to measure the electrochemical capacitance, CVs were obtained from 0 to 0.1 V (versus RHE) in 1 M KOH with sweep rates of 20, 40, 60, 80, and 100 mV $s^{-1}$. Electrochemical impedance spectroscopy (EIS) was performed at various overpotentials with frequency from 0.05 to $10^6$ Hz with an AC voltage of 5 mV. For the Faradaic efficiency measurements, gas chromatography (Shimadzu, GC-2010 Plus) equipped with a molecular sieve column (length: 30 m, inner diameter: 0.53 mm, film thickness: 50 µm) and BID detector was employed to determine the experimental amount of $H_2$ evolved during a 60 min electrocatalytic process. The theoretical $H_2$ generation value was calculated using Faraday's law, based on the experimentally determined i–t plot. Analysis of the Faradic efficiency determined the amount of $H_2$ generated using NiO@1T-$MoS_2$/CFP to be consistent with its theoretical value, with an average Faradic efficiency of ~99.5% (Supplementary Fig. 12).

## Data availability

The data underlying Figs. 3–5, Supplementary Figs. 6–8 and 17 are provided as a Source Data file. The other data that support the findings of this study are available from the corresponding author upon request.

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

## Acknowledgements

The authors thank the UC Irvine Materials Research Institute (IMRI) for helping with SEM, STEM, and XRD characterizations. IMRI was funded in part by the National Science Foundation Major Research Instrumentation Program under grant no. CHE-1338173. The authors thank the financial support from the National Natural Science Foundation of China (NSFC Nos. 91022010, 21631007, 21471087, and 21225103), the Specialized Research Fund for the Doctoral Program of Higher Education of China, and Tsinghua University Initiative Foundation Research Program (No. 20131089204). Jing Gu acknowledges San Diego State University (SDSU) start-up funds, the SDSU University Grants Program, and NSF award CEBT-1704992. Use of the Stanford Synchrotron Radiation Lightsource, SLAC National Accelerator Laboratory, is supported by the U.S. Department of Energy, Office of Science, and Office of Basic Energy Sciences under Contract No. DE-AC02-76SF00515. L.J.Z. acknowledges the support of the NSFC (Grant 61722403 and 11674121) and Program for JLU Science and Technology Innovative Research Team. Calculations were performed in part at High Performance Computing Center of Jilin University.

## Author contributions

Y.G.W. and J.G. led this research. Y.C.H., J.G. and Y.G.W. conceived the project and designed the experiments. Y.C.H., N.W., S.Y., J.H. and J.X.G. helped perform the synthetical experiments and characterizations. Y.C.H. and W.B. helped conduct electrochemical measurements. Y.H.S. and X.H. performed the first-principles energetic calculations. Y.H.S. and L.J.Z. analyzed the theoretical data. X.L.Z., N.P., P.B. and J.E.H. performed the XAS measurements and data fitting. T.A., X.X.Y. and X.Q.P. conducted the aberration-corrected high-resolution HAADF-STEM. Y.C.H., Y.H.S., L.J.Z., J.G. and Y.G.W. wrote the manuscript with inputs from all the authors.
