## [Peer Review File · Nature Communications]

Reviewers' comments:

Reviewer #1 (Remarks to the Author):

The revision manuscript is quite improved by additional useful data, such as STEM analysis and XAS. However, some key points in this work still need be further addressed.

1, for X-ray absorption spectroscopy test, the Mo K-edge is nice, but the detail analysis of Ni k-edge XAFS is missed that will be important to identify the position of atomic Ni dopant. The authors need to add this. Besides, the related figure captions should be carefully mentioned, such as the difference of XAS, XAFS, XANES and EXAFS, which element, which edge.

2, to claim the Oxygen chemical bonding, the author should also provide X-ray absorption spectroscopy data for further proving the critical bond of Ni-O-Mo.

3, as comparison in the Mo3d XPS of XO@1T-MoS₂, why upshift in Mo3d but down-shift in S 2p for Fe-doped samples? Does this mean the doped mechanism or Fe-doped position difference with the other two? The author need give more careful understanding on the real physical mechanism of upshift and downshift in XRS data.

4, to claim the stability of doped structure, the material's characterizations after cycling should be provided, such as Raman or X-ray absorption spectroscopy tests.

5, the authors mentioned that the introduction of secondary metal (Fe, Co, Ni) does not change the pristine 1T-MoS₂ structures in Fig. 4a, it is unusual. The phonons spectrum simulations via CASTEP package are needed to confirm this fact.

6, In Fig. 4c, the authors ascribed the big S₆₊ peak to the oxidation in air. Does the this also affect the HER performance? The authors need discuss this point.

Thus, I would suggest major revision before publish.

Reviewer #2 (Remarks to the Author):

Revised version has generally addressed major conceptual concerns. Particularly, the newly supplied HAADF STEM is now clearer and displays intensity modulation expected of Ni.

Further questions:

1.Lines in Figure 2e (HAADF-STEM) is not visible in print.

2.How large is the observation window for the HAADF-STEM? Is Ni observed in larger area or just at the zoomed-in area shown in Fig 2e?

3.The description for X-ray absorption spectroscopy needs to be improved. The authors refer Fig 3a as XANES (182-183), while Fig 3b is referred to as EXAFS (Line 191-192). Does that mean Figure 3b is derived from other spectra not shown?

4.Furthermore, experimental details on X-ray adsorption experiments and data treatment and fitting

procedures are missing. XAS methods (line 576-584) only describes collection of Mo K edge spectra, however other edges are actually shown (e.g. Figure S6). What is 3M tape? (Line 584). Some tapes absorb X-ray fluorescence and may contain metallic impurities.

5. The authors must improve the formatting and arrangement of supporting information (SI) more carefully. The SI looks very much disorganised. For example, there is additional information on EXAFS fitting on Table 2, but it is separated from Figure S6 and S7. There should be accompanying text on all supporting figures and tables to explain what they are and what are the significance of the data shown in relation to the main text.

6. Figure S6 description cannot be correct as Ni K-edge energy range should be around 8300 to 8700 eV.

7. The Ni K-edge EXAFS (Figure S6b) should be investigated in more detail. Fitting should be done to obtain at least the first two shells information, like what the authors did for the Mo edge. This is arguably more important than the fitting shown in Fig 3d and Fig S7, as this data can potentially corroborate the proposed structure model shown in Figure 1c, where Ni-S (or Ni-O) and Ni-Mo is expected in first and second shells respectively. Additionally, the Energy-intensity plot where both Ni@1T-MoS₂ and NiO standard where it is derived must be included in SI.

8. It is also interesting to see if the assignment of Ni-O bonding in Fig S6b (SI line 91-94) can be proven (as opposed to Ni-S), probably by systematic measurement of Ni K edge on T-160, T-180, T-200 and T-220 which the authors proposed to have decreasing O content.

9. The performance improvement of Ni@1T-MoS₂ in acidic electrolyte (Fig S17) is not as much as it is in alkaline electrolyte. In fact the onset potential and the Tafel slope of the Ni@1T-MoS₂ are very similar to undoped 1T-MoS₂. I think this needs to be explained. As a precaution the authors should test the alkaline KOH electrolyte for metallic contamination by ICP, before and after being used for HER.

10. Line 59-64, actually strategies to effectively improve catalyst active sites has been evaluated, including the use of non-noble metals and expand active sites beyond edges for 2D materials (see 10.1016/j.mattod.2018.01.034).

11. One point that is not answered properly from Rev 3 Comment 7 is the presence of S₆₊ even before cycling, which may indicate excess S or sulphate on surface.

Response to Reviewers

We sincerely appreciate the reviewers for spending their valuable time evaluating our manuscript and giving us constructive suggestions to improve the quality of the manuscript. The point-to-point responses to the reviewers' comments are attached below and all the corresponding revisions newly made are highlighted green in the revised manuscript.

Reviewer #1: *The revision manuscript is quite improved by additional useful data, such as STEM analysis and XAS. However, some key points in this work still need be further addressed. Thus, I would suggest major revision before publish.*

Response: We thank the reviewer for his/her positive comments on our work. The following comments have been addressed point-by-point.

Comment 1: *For X-ray absorption spectroscopy test, the Mo K-edge is nice, but the detail analysis of Ni k-edge XAFS is missed that will be important to identify the position of atomic Ni dopant. The authors need to add this. Besides, the related figure captions should be carefully mentioned, such as the difference of XAS, XAFS, XANES and EXAFS, which element, which edge.*

Response: We thank the reviewer for raising this point. We have carefully reformatted and reanalyzed all XAS data presented in the manuscript. The Ni K-edge results for NiO@1T-MoS₂ are now presented in the main text (see Figure 3c-d) and are properly labeled and discussed. Additionally, we provide the full energy-range EXAFS spectra in energy space, and any comparisons that will facilitate the reader's understanding within the SI. We strongly agree that it is crucial to investigate the local structure of the Ni dopant. XAS analysis showed that Ni is located in a similar S-rich octahedral coordination environment to Mo and the existence of Ni-Mo scattering, implying that it is successfully incorporated into the 1T-MoS₂ lattice.

Figure 3 | X-ray absorption analysis. **a.** The normalized Mo K-edge XANES spectra and first derivative inset. **b.** Comparison of R-space data and best fit lines. **c.** Ni K-edge XANES spectrum for NiO@1T-MoS₂ with enlarged pre-edge region inset. **d.** Corresponding EXAFS spectrum fitting of NiO@1T-MoS₂ in R-space with K-space inset.

Comment 2: To claim the Oxygen chemical bonding, the author should also provide X-ray absorption spectroscopy data for further proving the critical bond of Ni-O-Mo.

Response: Thanks for the suggestion. Two strategies are employed to address this question.

1) The Ni K-edge data for NiO@1T-MoS₂ was reanalyzed by placing a Ni atom in the same environment as a Mo atom, using the same model in Mo K-edge data (see Fig. 3d). In the first shell, it was only reasonable to include Ni-S paths, since any Ni-O paths are much smaller in intensity and, therefore, don't contribute to the signal enough to be considered in the model. In the second shell, it was found that the best fit results are obtained by including Ni-Mo single scattering path. However, the apparent coordination number for Ni-Mo fit to approximately 0.5, which is due to the large S atoms blocking secondshell scattering. It is conceivable that the small second shell scattering observed is through Ni-O-Mo bridging, and we suggest this as a possibility.

2) We performed an additional experiment of measuring the XAS of T-160, T-180, T-200, and T-220. Since the NiO@1T-MoS₂ samples prepared at different temperatures that showed lower O content with higher temperature (see Table S1), we were able to observe increased Ni-S coordination with increasing temperature, which implies lower Ni-O coordination (see Fig. S17a). We are unable to include Ni-O single scattering into the model because the scattering amplitude of Ni-S is prohibitively intense compared to Ni-O. Importantly, we observe Ni-Mo single scattering for all samples (see Fig. S17c), implying that changing the O incorporation does not result in differing second-shell structure. Together, we believe that these results show that Ni is incorporated into the 1T-MoS₂ lattice and that Ni-O-Mo bridging likely results in the observation of Ni-Mo scattering.

Figure S17. **a.** Ni K-edge EXAFS and **b.** XANES spectra for NiO@1T-MoS₂ synthesized at different temperatures, in Celsius degrees. Inset of **a** compares R-space spectra, inset of **b** compares first derivative spectra. **c.** The best fits in R-space and **d.** K-space with the corresponding data shown in Table S5. For the sake of comparison, the NiO@1T-MoS₂ data in Figure 3, main text, is replotted here (synthesis temperature of 180 Celsius degree).

Comment 3: As comparison in the Mo3d XPS of XO@1T-MoS₂, why upshift in Mo3d but down-shift in S 2p for Fe-doped samples? Does this mean the doped mechanism or Fe-doped position difference with the other two? The author need give more careful understanding on the real physical mechanism of upshift and downshift in XPS data.

Response: We appreciate the reviewer's comments for deepening our understanding of the doping effect. Based on Figure S8, both Mo3d and S2p of FeO@1T-MoS₂ show upshift compared to the analogue 1T-MoS₂ (see Fig. S8b and S8c). For the bare 1T-MoS₂ nanosheets, the peaks of Mo 3d_{5/2} and Mo 3d_{3/2} appear at 228.63 and 231.77 eV while the S 2p_{3/2} and S 2p_{1/2} peaks are located at 161.51 and 162.98 eV, respectively. For FeO@1T-MoS₂ nanosheets, the peaks of Mo 3d_{5/2} and Mo 3d_{3/2} slightly upshift to 228.71 and 231.90 eV, respectively. Simultaneously, S 2p_{3/2} and S 2p_{1/2} are also slightly upshifted to 161.64 and 163.08 eV, respectively. For NiO@1T-MoS₂ nanosheets, the peaks of Mo 3d_{5/2} and Mo 3d_{3/2} obviously upshift to 229.22 and 232.41 eV, respectively and the peaks of S 2p_{3/2} and S 2p_{1/2} also obviously upshift to 162.02 and 163.23 eV, respectively. In contrast, for CoO@1T-MoS₂ nanosheets, the peaks of Mo 3d_{5/2} and Mo 3d_{3/2} obviously downshift to 228.42 and 231.63 eV, respectively. Similarly, the peaks of S 2p_{3/2} and S 2p_{1/2} downshift to 161.23 and 162.44 eV, respectively. Due to the large amount present of S²⁻ and the corresponding Fe, Co and Ni XPS, Fe, Co, and Ni are mainly existing in the oxidation state of 2+. The dopant mechanism might be different in between Fe/Ni and Co. The upshift of the peaks is usually attributed to the n-doping process, which causes a Fermi level shift towards the conduction band edge or electron transfer from 1T-MoS₂ to dopant (Nano Lett. 2013, 13, 1991-1995). While the downshift is indicative of p-doping process, which indicates the reduction of 1T-MoS₂ due to the transfer of electrons from the dopants (Nanoscale, 2017, 9, 3576).

Figure S8b. High-resolution XPS signal of Mo 3d in XO@1T-MoS₂ (colour code: green: 1T-MoS₂; blue: FeO@1T-MoS₂; light blue: CoO@1T-MoS₂; purple: NiO@1T-MoS₂). **S8c.** High-resolution XPS signal of S 2p in XO@1T-MoS₂ (colour code: green: 1T-MoS₂; blue: FeO@1T-MoS₂; light blue: CoO@1T-MoS₂; purple: NiO@1T-MoS₂).

Comment 4: To claim the stability of doped structure, the material's characterizations after cycling should be provided, such as Raman or X-ray absorption spectroscopy tests.

Response: Thanks for the reviewer's comment. The Raman spectra of NiO@1T-MoS₂/CFP catalyst before and after 30 h long-term HER stability testing have been tested and provided in Fig.

S11a.

Figure S11. Characterization of NiO@1T-MoS₂/CFP after long-term HER stability testing. **a.** The Raman patterns of NiO@1T-MoS₂/CFP initial (black curve) and after 30 hours (blue curve) in 1.0 M KOH for HER. **b.** SEM image, **c.** TEM image, and **d.** EDS mappings of Mo, Ni, S, O elements in NiO@1T-MoS₂/CFP after long-term HER stability testing in 1.0 M KOH.

Comment 5: The authors mentioned that the introduction of secondary metal (Fe, Co, Ni) does not change the pristine 1T-MoS₂ structures in Fig. 4a, it is unusual. The phonons spectrum simulations via CASTEP package are needed to confirm this fact.

Response: We agree with the reviewer that substituting Mo in 1T-MoS₂ by the secondary metal X (Fe, Co, Ni) will lead to modification of vibration spectrum, since the changed X-S chemical bonding alters force constants and meanwhile the atomic mass of X is different. We have carefully reexamined the Raman spectra of all the fabricated X-doped 1T-MoS₂ samples. We found that the zoomed-in spectra in the region of 100-250 cm⁻¹ and 250-350 cm⁻¹ do show small peak shifts and peak intensity changes upon the secondary X metal doping, as shown in the updated Fig. 4a. Following the reviewer's suggestion, we calculated the local vibration frequencies associated with metal X and its nearest-neighbor S atoms with the first-principles code we used for catalytic activity simulation, VASP. The results indicate that the local vibration frequencies vary moderately from 95-393 cm⁻¹ of Mo@1T-MoS₂, 97-288 cm⁻¹ of Fe@1T-MoS₂, 104-303 cm⁻¹ of Co@1T-MoS₂, to 105-339 cm⁻¹ of Ni@1T-MoS₂. Meanwhile, the small amount of X (around 1/6) existing in the synthesized X@1T-MoS₂ samples is expected to result in a rather weak vibration intensity. These may be responsible for the observed small changes of the Raman spectra of X@1T-MoS₂ (X = Fe, Co, Ni) in Fig. 4a.

In response to the reviewer's comment, we have added a cautionary note to the manuscript clarifying this point: *We note that substituting Mo in 1T-MoS₂ by the secondary metal X with different chemical property and atomic mass (Fe, Co, Ni) will in principle lead to modification of vibration spectrum, as indeed indicated by our first-principles local vibration frequencies calculations. The moderate frequency shifts and the weak vibration intensity owing to the small amount of X may be responsible for the observed slight changes of Raman spectra of X@1T-MoS₂ (X = Fe, Co, Ni) (Fig. 4a and Fig. S20).*

Figure S20 Raman spectra at different ranges. **a.** Raman spectra of the XO@1T-MoS₂ nanosheet at a range of 100-250 cm⁻¹. **b.** Raman spectra of the XO@1T-MoS₂ nanosheet at a range of 250-350 cm⁻¹. (red: 2H-MoS₂; green: 1T-MoS₂; blue: FeO@1T-MoS₂; light blue: CoO@1T-MoS₂; pink: NiO@1T-MoS₂).

Comment 6: In Fig. 4c, the authors ascribed the big S6+ peak to the oxidation in air. Does this also affect the HER performance? The authors need discuss this point.

Response: We thank the reviewer for raising this point. Actually, we found out by further experiment that the S⁶⁺ peak might be derived from the sulfate moieties physically absorbed on the surface of planar 2D 1T-MoS₂ sheets, which can be easily removed by washed with deionized water (Figure S21a). Moreover, the removal of sulfate moieties has little influence on the HER performance of NiO@1T-MoS₂/CFP (Figure S21b).

Figure S21. The influence of S⁶⁺ (sulfate moieties) on HER performance. **a.** XPS spectra of S2p for the as-prepared NiO@1T-MoS₂/CFP and that after washing. **b.** The HER performance of NiO@1T-MoS₂/CFP and that after washing. Note: The as-prepared NiO@1T-MoS₂/CFP sample

is washed by 8 mL deionized water with shaking (repeat 10 times) to remove the sulfate moieties.

Reviewer #2: *Revised version has generally addressed major conceptual concerns. Particularly, the newly supplied HAADF STEM is now clearer and displays intensity modulation expected of Ni.*

Response: We appreciate the reviewer for his/her positive assessment to our work.

Comment 1: *Lines in Figure 2e (HAADF-STEM) is not visible in print.*

Response: The lines' colors in Fig. 2e have been corrected.

Comment 2: *How large is the observation window for the HAADF-STEM? Is Ni observed in larger area or just at the zoomed-in area shown in Fig 2e?*

Response: The size of observation window for HAADF-STEM in Fig 2e is about 2 nm * 2 nm. In larger area images, the stacking of NiO@1T-MoS₂ nanosheets result in overlap of the Ni and Mo atoms, making it almost impossible to distinguish between those two element. In order to show clear HAADF-STEM results that identify the Ni dopants in the 1T-MoS₂, we have to find out single layer area as shown in Fig. 2e.

Comment 3: *The description for X-ray absorption spectroscopy needs to be improved. The authors refer Fig 3a as XANES (182-183), while Fig 3b is referred to as EXAFS (Line 191-192). Does that mean Figure 3b is derived from other spectra not shown?*

Response: We thank the reviewer for this comment. We have carefully reformatted and reanalyzed all XAS data presented in the manuscript. Regarding the specific question that the reviewer raises, the spectra are all derived from the EXAFS data set in energy space. The original Figure 3a was highlighting the XANES portion of the EXAFS spectrum (simply by changing the x-axis bounds of the EXAFS chart). The original Figure 3b showed the entire above-edge region of the EXAFS spectrum reverse Fourier-transformed into K-space. The K-space data is then Fourier-transformed to produce R-space data. This is the typical data work-up for XAS, however we failed to show all steps of the process in the original manuscript. We have updated the SI to include the full EXAFS energy range spectrum in energy-space and to show clear comparisons of each type of spectrum to ensure that all data is presented. For Mo K-edge data, it is shown in Figure 3a-b and Figure S6. For Ni K-edge data, it is shown in Figure 3c-d, Figure S7 and S17.

Original Figure 3 | X-ray absorption analysis of Mo's K-edge in the NiO@1T-MoS₂ catalyst. **a.** The normalized Mo K-edge XANES spectra. **b.** Mo's K-edge EXAFS oscillation functions $\kappa^3\chi(k)$. **c.** Corresponding Fourier transformation (FT) of the Mo's K-edge EXAFS spectra analyses, which reveal the shortening in the Mo-Mo bond length within the NiO@1T-MoS₂ nanosheets compared with the bulk 2H-MoS₂ sample. **d.** Corresponding Mo's K-edge EXAFS spectrum fitting of NiO@1T-MoS₂ at R space.

Figure 3 | X-ray absorption analysis. **a.** The normalized Mo K-edge XANES spectra and first derivative inset. **b.** Comparison of R-space data and best fit lines. **c.** Ni K-edge XANES spectrum for NiO@1T-MoS₂ with enlarged pre-edge region inset. **d.** Corresponding EXAFS spectrum fitting of NiO@1T-MoS₂ in R-space with K-space inset.

Figure S6. The normalized Mo K-edge EXAFS (a) and XANES (b) spectra of 2H-MoS₂, 1T-MoS₂, and XO@1T-MoS₂. The first derivative inset of (b) compares the XANES edge position, where it is observed that Ni has the most pronounced shift. EXAFS fitting results in R-space (c) and K-space (d). For clarity in comparison, the 2H-MoS₂ and NiO@1T-MoS₂ data from Figure 3, main text, is replotted here. The corresponding fit data is presented in Table S2.

Figure S7. Comparison of the Ni K-edge XAS spectra in the EXAFS (a) and XANES (b) energy ranges. The K-space (c) and R-space (d) comparisons with data shown in open points and best fits in solid lines. The corresponding NiO reference fit parameters are shown in Table S3.

Figure S17. Ni K-edge EXAFS (a) and XANES (b) spectra for NiO@1T-MoS₂ synthesized at different temperatures, in Celsius degrees. Inset of (a) compares R-space spectra, inset of (b) compares first derivative spectra. The best fits in R-space (c) and K-space (d) with the corresponding data shown in Table S5. For the sake of comparison, the NiO@1T-MoS₂ data in Figure 3, main text, is replotted here (synthesis temperature of 180 degrees Celsius).

Comment 4: Furthermore, experimental details on X-ray adsorption experiments and data treatment and fitting procedures are missing. XAS methods (line 576-584) only describes collection of Mo K edge spectra, however other edges are actually shown (e.g. Figure S6). What is 3M tape? (Line 584). Some tapes absorbs X-ray fluorescence and may contain metallic impurities.

Response: Thanks for comments. The experimental details on X-ray adsorption experiments and data treatment and fitting procedures are improved and shown in XAS part. The 3M tape is Kapton polyimide film, doesn't contain any metal that would influence the X-ray adsorption experiments, purchased from 3M.

Comment 5: The authors must improve the formatting and arrangement of supporting information (SI) more carefully. The SI looks very much disorganised. For example, there is additional information on EXAFS fitting on Table S2, but it is separated from Figure S6 and S7. There should be accompanying text on all supporting figures and tables to explain what they are and what are the significance of the data shown in relation to the main text.

Response: Thanks for comments. The supporting information (SI) has been reorganized and formatted according to the reviewer's suggestions. All the supporting figures and tables about EXAFS fitting have been accompanying text to explain.

Comment 6: Figure S6 description cannot be correct as Ni K-edge energy range should be around 8300 to 8700 eV.

Response: Thanks for comments. We have re-analyzed and updated all the XAS plots, especially the Figure S6.

Comment 7: The Ni K-edge EXAFS (Figure S6b) should be investigated in more detail. Fitting should be done to obtain at least the first two shells information, like what the authors did for the Mo edge. This is arguably more important than the fitting shown in Fig 3d and Fig S7, as this data can potentially corroborate the proposed structure model shown in Figure 1c, where Ni-S (or Ni-O) and Ni-Mo is expected in first and second shells respectively. Additionally, the Energy-intensity plot where both Ni@1T-MoS₂ and NiO standard where it is derived must be included in SI.

Response: Thanks for the suggestions. The reanalysis XAS data includes multiple fitting models as recommended by the reviewer. The Ni K-edge data for NiO@1T-MoS₂ was reanalyzed by placing a Ni atom in the same environment as a Mo atom, using the same model in Mo K-edge data. In the first shell, it was only reasonable to include Ni-S paths, since any Ni-O paths are much smaller in intensity and, therefore, don't contribute to the signal enough to be considered in the model. In the second shell, it was found that the best fit results are obtained by including Ni-Mo single scattering path. However, the apparent coordination number for Ni-Mo fit to approximately 0.5, which is due to the large S atoms blocking secondshell scattering. It is conceivable that the small second shell scattering observed is through Ni-O-Mo bridging, and we suggest this as a possibility. Regarding the Ni@1T-MoS₂ and NiO standard comparison, we agree that the previous figure was insufficient. We have clearly reformatted the data to show this comparison in full EXAFS energy range, XANES region, K-space, and R-space (Figure S7).

Figure S7. Comparison of the Ni K-edge XAS spectra in the EXAFS (a) and XANES (b) energy ranges. The K-space (c) and R-space (d) comparisons with data shown in open points and best fits in solid lines. The corresponding NiO reference fit parameters are shown in Table S3.

Comment 8: It is also interesting to see if the assignment of Ni-O bonding in Fig S6b (SI line 91-94) can be proven (as opposed to Ni-S), probably by systematic measurement of Ni K edge on T-160, T-180, T-200 and T-220 which the authors proposed to have decreasing O content.

Response: This suggestion intrigued us to perform the following experiment shown in Figure S17 and Table S5. The Fourier-transformed R-space spectra comparison in Figure S17a suggest that higher synthesis temperatures result in increased coordination number of Ni-S (assuming that the Debye-Waller factor, particularly its static disorder contribution, remains similar for all samples). Indeed, quantitative results (Figure S17c-d and Table S5) indicate that the degree of Ni-S coordination is increased with increasing synthesis temperature. This result is in agreement with ICP-MS results that showed lower O incorporation at higher temperature. Unfortunately, for the reason mentioned in comment 7 response, we cannot include Ni-O in the model. Importantly, we observe Ni-Mo single scattering for all of the samples, implying that changing the O incorporation does not result in differing second-shell structure. However, due to the low apparent coordination number of Ni-Mo, we are unable to provide a quantitative trend of Ni-Mo coordination as a function of O content. Otherwise, we speculate that it might have been possible to assess Ni-O-Mo by intensity of Ni-Mo since the smaller O ligand could allow more efficient second-shell scattering. Despite this limitation, we believe that the clear observation of Ni-Mo single scattering, combined with XPS results, suggests the presence of Ni-O-Mo moieties.

Figure S17. Ni K-edge EXAFS (a) and XANES (b) spectra for NiO@1T-MoS₂ synthesized at different temperatures, in Celsius degrees. Inset of (a) compares R-space spectra, inset of (b) compares first derivative spectra. The best fits in R-space (c) and K-space (d) with the corresponding data shown in Table S5. For the sake of comparison, the NiO@1T-MoS₂ data in Figure 3, main text, is replotted here (synthesis temperature of 180 Celsius degree).

Table S5. Summary of local structural parameters in NiO@1T-MoS₂ synthesized at different

temperatures, fitted from Ni K-edge EXAFS data

Temp.	Ni-S Single Scatter				Ni-Mo Single Scatter			
	E_0	σ^2 (\AA^2)	N	R (\AA)	E_0	σ^2 (\AA^2)	N	R (\AA)
160	-10.2	0.008	3.99	2.20	-6.5	0.01	0.56	3.10
180	-8.5	0.008	4.27	2.21	-10	0.01	0.40	3.05
200	-7.0	0.008	4.51	2.22	-10	0.01	0.42	3.14
220	-2.0	0.008	4.90	2.23	-10	0.01	0.50	3.11

The structural parameters around Ni atoms of NiO@1T-MoS₂ as a function of synthesis temperature were fit using the same model utilized at Mo K-edge, by replacing Mo with Ni. Increased synthesis temperature results in an increased coordination number of S. Due to the relatively small scattering intensity of O, Ni-O paths could not be reasonably incorporated into the fitting model. Due to the presence of O, Ni-Mo paths could be included. While the expected highest value of N is observed in T-160, the sample with the highest O content, there is no strong trend due to large uncertainty in N value. However, the clear observation of Ni-Mo single scattering in all samples suggests the presence of Ni-O-Mo moieties.

Note: R is the length of the vector, N is the coordination number, σ^2 is Debye-Waller factor and ΔE is the edge-energy shift. The obtained R values are approximately ± 0.02 \AA .

Comment 9: The performance improvement of Ni@1T-MoS₂ in acidic electrolyte (Fig S17) is not as much as it is in alkaline electrolyte. In fact, the onset potential and the Tafel slope of the Ni@1T-MoS₂ are very similar to undoped 1T-MoS₂. I think this needs to be explained. As a precaution the authors should test the alkaline KOH electrolyte for metallic contamination by ICP, before and after being used for HER.

Response: We appreciate the reviewer for making constructive comments. The different performance of the catalysts under acid and basic condition might be due to different mechanisms for HER at different pHs. Our DFT calculation results (Figure 6) shows the oxygen and Ni incorporations in NiO@1T-MoS₂ nanosheets will greatly lower the free energy for the water dissociation step and lower the free energy for the H^{*} desorption process. In an alkaline condition, the rate-determining step is water dissociation step, the free energy change of this step can greatly improve the HER kinetic. However, suppressing of the H^{*} desorption energy to be too low might reduce the HER kinetics, especially in an acidic electrolyte while the desorption of H^{*} might be the rate determine step. That is why NiO@1T-MoS₂ nanosheets does not show much catalytic improvement under acidic solution.

In a contrary, for the control 1T-MoS₂ (light blue), the free energy of the H^{*} desorption process is high thus the oxygen incorporations into the control 1T-MoS₂ will improve the HER kinetics, especially in an acidic electrolyte. While for the NiO@1T-MoS₂, the integrated incorporations of Ni and O atoms make the HER performance similar to that of 1T-MoS₂ in an acidic electrolyte as shown in Fig. S18.

Meanwhile, as the reviewer suggested, we have analyzed the alkaline KOH electrolyte before and after HER by ICP and didn't find any metallic contamination.

Figure 6 | First-principles calculations of the effect on HER performance of various doped transition metals. **a.** The monolayer structure of X@1T-MoS₂ formed by the XMo₆ hexagon motifs. **b.** Free energy diagrams on the surface of different catalysts in alkaline solution, with the smallest repeating unit cell used in calculations indicated. **c.** The structures of predicted intermediates that bind to water and hydrogen. The primitive cell is labeled with light blue frame and cyan background (top left). The green, yellow, and orange spheres represent Mo, S, and X (Fe, Co, Ni) atoms, respectively.

Comment 10: Line 59-64, actually strategies to effectively improve catalyst active sites has been evaluated, including the use of non-noble metals and expand active sites beyond edges for 2D materials (see 10.1016/j.mattod.2018.01.034).

Response: Thanks for the reviewer's reminder. The review that published in Materials Today (DOI: 10.1016/j.mattod.2018.01.034) well summarized the advancements for ultrathin 2D materials in catalytic hydrogen evolutions applications. Some strategies to effectively improve catalyst active sites including chemical doping and phase changing, vacancy, or disorder engineering in ultrathin 2D materials are tightly connected with this manuscript, thus has been cited as Ref. 30 in the revised manuscript.

Comment 11: One point that is not answered properly from Rev 3 Comment 7 is the presence of S₆⁺ even before cycling, which may indicate excess S or sulphate on surface.

Response: We thank the reviewer for raising this point. Our further experimental results show that the big S₆⁺ peak is derived from the sulfate moieties physically absorbed on the surface of planar 2D 1T-MoS₂ sheets, which can be easily removed by washed with deionized water (Figure S20a). Moreover, the removal of sulfate moieties has little influence on the HER performance of NiO@1T-MoS₂/CFP (Figure S20b).

Figure S20. The influence of $S6+$ (sulfate moieties) on HER performance. **a.** XPS spectra of S2p for the as-prepared NiO@1T-MoS₂/CFP and that after washing. **b.** The HER performance of NiO@1T-MoS₂/CFP and that after washing. Note: The as-prepared NiO@1T-MoS₂/CFP sample was washed with 8 mL deionized water and shaken (repeat 10 times) to remove the sulfate moieties.

REVIEWERS' COMMENTS:

Reviewer #1 (Remarks to the Author):

The concerns on structures have been addressed, the revision is acceptance for publishing.

Reviewer #2 (Remarks to the Author):

The authors have answered the questions raised and publication of manuscript is recommended.

Response to Reviewers

We sincerely appreciate the reviewers for spending their valuable time evaluating our manuscript and giving us constructive suggestions to improve the quality of the manuscript. The point-to-point responses to the reviewers' comments are attached below and all the corresponding revisions newly made are highlighted green in the revised manuscript.

REVIEWERS' COMMENTS:**Reviewer #1** (Remarks to the Author):

The concerns on structures have been addressed, the revision is acceptance for publishing.

Reviewer #2 (Remarks to the Author):

The authors have answered the questions raised and publication of manuscript is recommended.